# Maternal Melatonin Deficiency Leads to Endocrine Pathologies in Children in Early Ontogenesis

**DOI:** 10.3390/ijms22042058

**Published:** 2021-02-19

**Authors:** Dmitry O. Ivanov, Inna I. Evsyukova, Ekaterina S. Mironova, Victoria O. Polyakova, Igor M. Kvetnoy, Ruslan A. Nasyrov

**Affiliations:** 1Department of Neonatology, Saint-Petersburg State Pediatric Medical University, Litovskaya Ulitsa, 2, St. Petersburg 194100, Russia; doivanov@yandex.ru (D.O.I.); vopol@yandex.ru (V.O.P.); rrmd99@mail.ru (R.A.N.); 2Ott Research Institute of Obstetrics, Gynecology and Reproductology, Mendeleyevskaya Liniya, 3, St. Petersburg 199034, Russia; eevs@yandex.ru; 3Saint Petersburg Institute of Bioregulation and Gerontology, Dynamo Ave., 3, St. Petersburg 197110, Russia; 4Saint-Petersburg Research Institute of Phthisiopulmonology, Lygovsky Ave., 2-4, St. Petersburg 191036, Russia; igor.kvetnoy@yandex.ru; 5Department of Pathology, Saint-Petersburg State University, University Embankment, 7/9, St. Petersburg 199034, Russia

**Keywords:** melatonin, endocrine pathology, children, early ontogenesis, family planning

## Abstract

The review summarizes the results of experimental and clinical studies aimed at elucidating the causes and pathophysiological mechanisms of the development of endocrine pathology in children. The modern data on the role of epigenetic influences in the early ontogenesis of unfavorable factors that violate the patterns of the formation of regulatory mechanisms during periods of critical development of fetal organs and systems and contribute to the delayed development of pathological conditions are considered. The mechanisms of the participation of melatonin in the regulation of metabolic processes and the key role of maternal melatonin in the formation of the circadian system of regulation in the fetus and in the protection of the genetic program of its morphofunctional development during pregnancy complications are presented. Melatonin, by controlling DNA methylation and histone modification, prevents changes in gene expression that are directly related to the programming of endocrine pathology in offspring. Deficiency and absence of the circadian rhythm of maternal melatonin underlies violations of the genetic program for the development of hormonal and metabolic regulatory mechanisms of the functional systems of the child, which determines the programming and implementation of endocrine pathology in early ontogenesis, contributing to its development in later life. The significance of this factor in the pathophysiological mechanisms of endocrine disorders determines a new approach to risk assessment and timely prevention of offspring diseases even at the stage of family planning.

## 1. Introduction

In recent decades, we have seen an increase in endocrine disorders among children and adolescents, which predetermines the annual growth of incidence rates of serious chronic diseases among adults [1]. Thus, the number of obese children and adolescents over the last 40 years has increased from 11 million to 124 million, and there are 216 million overweight children [2]. This epidemic has mostly spread across developed countries in Europe and North America, yet there has been data showing a substantial growth of this pathology in developing countries [3,4]. Over the last decade, in Europe, the overweight (including obesity) rate in boys has peaked to 43% [5]. The metabolic syndrome occurs in 38% and 49.7% of overweight and obese children and adolescents, respectively [6]. By using eight criteria for diagnosing this metabolic syndrome, the authors note its incidence rate in the U.S. increasing from 6% to 39% among children of 4 to 16 years old [7] and up to 40% among people over 60 years old [8]. It is expected that by 2025 the metabolic syndrome will be diagnosed in 40% of the population in developing countries [9]. There is a concern about an early onset of type 2 diabetes in children, with rapid development of diabetic complications and cardiovascular pathologies [10,11,12]. Special attention should be given to increasing numbers of macrosomic newborns and the development of visceral obesity in children during the first months of life [13,14,15,16]. It is already in early infancy that various functional system disorders (gastrointestinal, cardiovascular, neural, etc.) and type 2 diabetes are diagnosed in such children [17,18,19,20]. Therefore, researchers are focused on studying the causes and pathogenic mechanisms of endocrine pathology development in children in early ontogenesis.

## 2. Causes of Postnatal Endocrine Pathologies in Offspring

Some researchers consider the problem of obesity in view of genetic predisposition to this disease [21]. Thus, they have identified eight monogenous and 50 syndromal forms of obesity, and the 12th version of the Human Obesity Gene Map contains over 600 genetic markers associated with the obesity phenotype. However, while a population’s genetic load remains the same, the obesity rates are rising [22].

A high probability of postnatal endocrine pathology development in offspring is currently associated with the epigenetic impact of adverse factors in early ontogenesis, which disrupt the formation of regulatory mechanisms in periods of critical development of the fetus’s organs and systems and contribute to the delayed development of pathological states, which take place in the case of maternal obesity and diabetes during pregnancy [23,24,25].

Researchers consider the effects of chronic bacterial and viral infections during pregnancy and factors such as stress, glucocorticoid therapy, eating habits, lifestyle and social and economic factors [26,27,28,29,30]. Special attention has been drawn to studying adverse consequences of gestational diabetes mellitus which complicates the course of pregnancy in overweight and obese women [31,32], with their share among women of reproductive age having increased to 70% in the USA and to 20–27% in Europe [33].

Based on clinical and experimental studies, various hypotheses have been proposed to explain the aetiopathogenesis of visceral obesity in the first months of life and its consequences both in overweight and underweight or premature newborns [34,35]. Some researchers argue that maternal undernutrition and somatic comorbidity leading to chronic placental insufficiency form the so-called fetal “thrifty phenotype” (“thrifty phenotype hypothesis”) which is responsible for subsequent building a strategy of survival, which is then realized through metabolic disorders and obesity development, type 2 diabetes and arterial hypertension [36].

It has been observed that the maternal low-protein diet in pregnancy results in offspring with expressed changes in the pancreas gland, a higher apoptosis rate and lower β-cell proliferation, a disrupted vascularization of Langerhans islets and a lower insulin content, which then leads to the development of type 2 diabetes [37,38]. Other hypotheses address the role of diabetes, metabolic syndrome, obesity, excessive nutrition and a high-fatty diet during pregnancy in the development of hyperglycemia, hyperinsulinemia, hyperleptinemia and increased cortisol levels in the fetus followed by a metabolic response modulation of hypothalamic neurons. Newborn infants had macrosomia, which was later followed by the development of obesity [39,40,41,42].

It is believed that excessive protein consumption in the early postnatal period (“Early Protein Hypothesis”) leads to a higher concentration of insulinogenic amino acids in the blood plasma which stimulate the production of insulin-like growth factor and insulin, which results in an excessive accumulation of adipose tissue and raises the risk of obesity. The absence of breastfeeding and higher protein levels in artificial feeding are viewed as risk factors for obesity development [43]. In recent decades, researchers’ attention has been drawn to studying the role of intestinal microbiota in the development of obesity.

It has been found that in obese pregnant women the content of intestinal microbiota shifts to more bacteria with anti-inflammatory properties (obesity is known to be followed by chronic inflammation). Intestinal microbiota imbalance leads to a higher concentration of short-chain fatty acids which act as signaling molecules and stimulate a cascade of reactions resulting in increased lipogenesis and stored fat [44,45]. Another mechanism is a lower expression of protein (a lipoprotein lipase inhibitor) in the case of dysbiosis.

This results in higher lipase activity, fatty acid uptake and storage in adipose tissue and muscles, which leads to increased stored fat in the organism [46,47]. Some studies have investigated the mechanism that consists of increased permeability of the intestinal barrier for gut microbiota endotoxin which binds with Toll-like receptors and triggers a cascade of reactions, followed by higher production of anti-inflammatory cytokines (TNF-α and IL-6) involved in the development of obesity and insulin resistance [48,49].

## 3. Melatonin and Regulation of Metabolic Processes

The diversity of etiological factors leading to obesity has drawn researchers’ attention to finding a key element that triggers this process. Numerous experimental and clinical studies have established the primary role of disruption of circadian rhythms of metabolic processes in the development of obesity in adults [50,51,52].

The integration between a cyclic environment and the circadian distribution of physiological and behavioral processes is provided by melatonin whose production circadian rhythms control all endogenous rhythms at tissue, cell and subcell levels [53,54,55]. This provides synchronization of metabolic processes, a capability to react to endogenous and exogenous impacts and the maintenance of stability and health of the organism [56].

Melatonin is synthesized in the pineal gland whose endocrine function is physiologically controlled by light. The photic information from retinal ganglion cells passes through the retinohypothalamic tract to the suprachiasmatic nuclei (SCN) of the hypothalamus, which are the circadian-rhythm generators or the biological clock. From there, the signals go to the superior cervical ganglia and then reach the pineal gland via sympathetic noradrenergic nerves. The activation of the beta-adrenergic receptors of pinealocytes triggers the synthesis of melatonin from serotonin with N-acetyltransferase (NAT) and hydroxyindole O-methyltransferase (HIOMT) enzymes.

Light suppresses melatonin production and secretion, and therefore its maximum level in human blood is observed at night and its minimum during the daytime [57,58,59]. Melatonin is not only synthesized in the pineal gland. Extrapineal melatonin is present in almost all organs: the gastrointestinal tract, liver, kidney, adrenal gland, heart, thymus, genital gland, placenta and uterus, as well as in platelets, eosinophiles, leukocytes and other immune system cells [60,61]. Melatonin synthesis in the mitochondria of eukaryotes suggests that melatonin protects cellular organelles against oxidative injury and maintains their physiological function [62].

Melatonin, through passive diffusion from pineal cells, passes to the cerebrospinal fluid and blood flow, thereby triggering chronobiological effects through binding with its receptors in cell membranes of various tissues. The circulating melatonin level reaches its peak at night-time (2–4 a.m.) and its minimum in the morning, with its values in the blood serum during night hours being 5–10 times higher.

Being hydrophilic, the melatonin molecule at the same time is highly lipophilic and therefore can easily pass through the blood-brain and placental barriers, reaching capillaries where 70% of melatonin binds to albumin. Melatonin’s half-life is about 30 to 45 min. Melatonin is metabolized in the liver and kidney, as well as in tissues (brain and intestine) where the enzymes involved in the process have been found [63]. The metabolism end-products are 6-sulfatoxymelatonin and acetylsalicylic acid.

Melatonin has a regulating effect through binding to receptors. Two types of membrane receptors (MT1 and MT2) and their localization in chromosomes (4q35 and 11q21-22), as well as nuclear receptors (RORα), have been identified [64]. It is thought through these receptors that melatonin triggers clock gene expression [65].

Circadian genes (*Clock*/*Bmal*) participate in the regulation of glucose and triglycerides levels at daytime [66], and *Bmal* regulates the synthesis of lipids and adipogenesis [67]. Melatonin receptors have been found in hypothalamic suprachiasmatic nuclei, cerebellum, retina, spleen, liver, pancreas, genital and mammary glands, uterus, thymus, gastrointestinal tract, thrombocytes, lymphocytes and adipose tissue [68].

Melatonin is also known as a key regulator of the carbohydrate and adipose metabolism. Its insulin-like hypoglycaemic, anabolic and anticholesterol effects have been demonstrated [69,70]. Numerous experimental molecular studies have demonstrated the mechanisms of functional relationships between melatonin, insulin and glucagon [71,72]. Furthermore, in the pancreas, as well as in other tissues, there are autonomous circadian genes (*Bmal*, *Clock*, *Per1*, *Cry1*), through which epiphyseal melatonin’s circadian rhythm provides a synchronizing effect on β- and α-cells [73,74]. Higher melatonin production and lower insulin secretion are observed at night, which is reversed at daytime [75,76].

The circadian amplitude of melatonin levels goes down in the case of night work and an active lifestyle, as well as in sleep disruptions, which leads to hyperinsulinemia, the development of insulin resistance and hyperleptinemia, a set of symptoms which are characteristic for type 2 diabetes [77,78,79,80]. The development of these pathological conditions related to low melatonin production and absence of its circadian rhythm is associated with distorted regulation of insulin synthesis and secretion by pancreas β-cells and its binding to target cell membrane receptors, as well as with suppression of GLUT4 and protein content reduction.

An experiment has shown that elimination of the pineal gland in insulin-sensitive tissues (white and brown adipose tissue and skeletal and cardiac muscles) leads to a dramatic drop in GLUT4 mRNA and microsomal and membrane proteins. In addition, in the absence of melatonin, the function of insulin receptors in adipocytes is disrupted, which sharply reduces the uptake of glucose by these cells. The identified effects of pinealectomy disappear after melatonin therapy [81].

In adipose tissue, melatonin regulates the differentiation of adipocytes and the expression of clock genes (*Clock*, *Per1*) there, increasing the expression of their mRNA at night [82]. Furthermore, melatonin entrains the circadian rhythms of lipogenesis, lipolysis and fatty acid and glucose uptake, thereby regulating lipidemia and glycemia, while at the same time supporting the circadian metabolism in muscles, liver and pancreas [73,83]. It is at night that the organism spends significant amounts of energy on supporting its functional systems’ vital activities [71].

It has been proved that long-time staying under continuous artificial illumination (which is characteristic for human activity today) suppresses the physiological rhythm of the pineal gland, which results in distorted interactions between melatonin and its receptors, leading to the development of insulin resistance and then to abdominal obesity, arterial hypertension and lipid and carbohydrate metabolism disorder [84,85,86].

## 4. The Role of Maternal Melatonin in the Formation of Circadian System in the Fetus

It has been shown that during pregnancy the circadian oscillations of melatonin levels substantially increase, reaching their peak before birth [87]. At the same time, the extrapineal melatonin production increases, especially in the placenta, which ensures the morphological development and stability of placental function and determines optimal adaptation to pregnancy and preparation for childbirth [88]. Additionally, in the placenta, since the earliest stages of pregnancy, there appears to be receptors of melatonin (MT1 and MT2) that enable the cyclic influence of maternal epiphyseal melatonin which activates and synchronizes the rhythmic expression of clock genes in trophoblast cells and endothelial tissue [87]. The circadian rhythm of clock gene activity in the placenta, coinciding with similar clock gene expression in the fetus’s liver, determines the development of fetal adaptive metabolic processes and normal growth [89,90].

Maternal epiphyseal melatonin plays an important role in the formation of circadian rhythms of clock genes involved in the regulation of metabolic processes and vital activities of fetal functional systems. The cyclic variations of melatonin levels coincide in maternal blood and fetal circulation [91]. The rhythmic expression of *Per1*, *Per2*, *Cry1* and *Bmal1* genes is clearly identifiable in fetal hypothalamic suprachiasmatic nuclei already in the middle of gestation. Even before birth, mRNA rhythms are identified there for vasopressin.

Furthermore, nerve fibers of the retinohypothalamic tract are also well identifiable during the second half of gestation [92,93,94]. It is assumed that the circadian regulation system starts functioning in the fetus with the appearance of receptors to melatonin being transferred from the mother since the earliest stages of pregnancy, providing information on the time of day [84,95,96]. A study of clock genes in a primate fetus demonstrated their circadian expression of varying degrees in the suprachiasmatic nuclei, adrenal gland, thyroid gland, brown adipose tissue and pineal gland, pointing to the role of maternal melatonin circadian signaling in the regulation of cortisol production by the fetal adrenal gland [97].

The circadian rhythm of melatonin secretion in pregnant rats was suppressed by exposure to constant light, which disrupted the clock gene rhythmic expression in all organs of the fetus, and it was reversed only after circadian injections of maternal melatonin [98]. The absence of melatonin secretion circadian rhythms in pregnant rats led to disruption of rhythmic clock gene expression in all sections of the gastrointestinal tract, liver and pancreas [99]. It has been shown that maternal melatonin synchronizes peripheral circadian oscillators in these organs and coordinates their function with clock gene rhythms of suprachiasmatic nuclei and other tissues, including the adenohypophysis and adrenal gland [100,101].

Circadian rhythms of clock gene expression in the fetal large intestine can be determined already by week 33 of intrauterine growth [102]. Furthermore, throughout the antenatal period, maternal melatonin is a key molecule that directs and coordinates the genetic process of developing relationships between clock genes of the infant’s organism and forming microbiota [103].

The melatonin synthesis in the fetal pineal gland is controlled by maternal melatonin during the fetus antenatal life [104], thus ensuring the genetic process of morphological and functional development of SCN and epiphysis [105,106]. It has been found that the melatonin rhythm in the maternal blood plasma triggers the fetal circadian system which, in turn, informs other tissues through corticosterone rhythmic signaling. Already in the last 10 weeks of pregnancy, fetal circadian rhythms of the cardiac rate and temperature are synchronized with the melatonin and cortisol content in maternal blood [107]. In absence of the melatonin circadian rhythm in the maternal blood plasma in the second half of pregnancy, there is intrauterine growth retardation, accompanied by delayed development of the circadian activity of the central rhythm driver, hypothalamic suprachiasmatic nuclei [59], changing the clock gene expression levels and disrupting the corticosterone production rhythm [108].

Based on results of experimental and clinical studies, a hypothesis has been proposed, according to which during antenatal life the fetal suprachiasmatic nuclei and organs are peripheral circadian oscillators whose rhythmic activity is triggered by and depends on the state of circadian organization of vital activities in the mother and its primary messenger of biorhythms generated by suprachiasmatic nuclei, i.e., melatonin. This ensures the postnatal integration of endogenous biorhythms of the baby’s functional systems into the adult-like circadian system which is regulated by its own suprachiasmatic nuclei in dependence to circadian changes in environment illumination [109].

In normal conditions, the formation of the circadian rhythm of epiphyseal melatonin production goes on rapidly in the first days and weeks of life, while the maternal impact on this process is exercised through breastfeeding as maternal milk contains high levels of melatonin with circadian oscillations [110]. The content of fat and the energy potential of breast milk are dependent on the circadian rhythms of melatonin production [111,112]. At night, breast milk has the highest content of components that, along with other effects, contribute to circadian-rhythm formation. For example, tryptophan and omega-3 polyunsaturated fatty acids inhibit monoamine oxidase and thereby increase the quantity of serotonin (a melatonin precursor) [113,114].

Therefore, in the case of breastfeeding, by the end of the perinatal period, the amount and daily rhythm of melatonin production in children correspond to that in adults [115]. The absence of circadian melatonin production during pregnancy impairs the genetic development of the rhythmic activity of suprachiasmatic nuclei and melatonin production in the fetal pineal gland, which leads to dysregulation of biochemical processes in the organism before and after birth [116,117].

Experimental studies have shown that, in those situations, there is also low epiphyseal melatonin production in offspring, and its circadian rhythm is absent not only at birth but in the postnatal period as well, which predetermines a mechanism of metabolic programming [55,118,119].

Our retrospective study regarding the occurrence of obesity by the end of the first year in 120 infants born from healthy mothers without pregnancy and delivery complications demonstrated that their body weight and length corresponded to normal levels, and there were no gastrointestinal dysfunctions. All the infants were breastfed for a period of 10–12 months and received complimentary feeding starting from 4–5 months. On the other hand, the body weight of 40 infants, born from mothers with obesity and pregnancy complicated by diabetes, exceeded 90–95% in 12 months after birth. They received maternal milk and feed formula for the first 3–4 months only, and complimentary feeding and milk formula from months 5–6 until the end of the first year, and they all had gastrointestinal dysfunctions.

The study of melatonin levels in maternal blood before delivery and in their newborn babies on the third day at day (11–12 a.m.) and night (4–5 a.m.) showed that the infants had melatonin production circadian rhythms if they were present in healthy mothers, although the total level was still lower: 1.9/11.5 and 12.2/81.6 pg/mL (day/night). On the other hand, they were absent in babies born from mothers with obesity and gestational diabetes: 3.6/5.2 and 31.9/35.7 pg/mL, respectively [120]. The excessive body weight and obesity in 6-month infants from diseased mothers suggests that the absence of melatonin production circadian rhythms is associated with an imbalance of energy exchange in early ontogenesis, which leads to the development of obesity.

Thus, in prenatal ontogenesis, maternal melatonin is a key molecule that directs and coordinates the genetic process of the child’s morphofunctional development, which is crucial for successful postnatal adaptation to a new environment and for healthy life in subsequent months and years.

## 5. The State of the Placenta and Fetal Development in the Case of Disruption of Maternal Melatonin Production

It is known that a genetic program is executed in interaction with environment factors, of which the most unfavorable are qualitative and quantitative malnutrition, smoking, narcotics and medications, chronic diseases, obesity, diabetes mellitus and gestational complications (preeclampsia, gestational diabetes, etc.). A complex of pathological processes arising during adaptation to pregnancy has a negative effect on the morphofunctional development of the placenta [121]. The activation of free-radical oxidation, depletion of antioxidant reserves and accumulation of peroxynitrite lead to oxidative stress [122,123,124,125,126]. Protein oxidative modification and lipid peroxidation result in dysfunction of cell membranes, receptors, enzymes, intracellular structures, particularly the endoplasmic reticulum (ER), and endothelial dysfunction [127,128,129]. The endoplasmic reticulum damage in endothelial cells is exacerbated by oxidation of triglycerides and low-density lipoproteins [130,131]. Endoplasmic reticulum oxidative stress leads to activation of inflammatory responses, inducing the secretion of interleukins (IL-1α, IL-1β), TNF-a and systemic inflammation [132,133].

This may result in disruption of uteroplacental circulation, hormone-producing functions of the placenta and its circadian rhythm, ischemia, chronic placental insufficiency and intrauterine growth retardation [134]. Numerous experimental studies have shown that the development of pathological processes is primarily due to insufficient production of epiphyseal melatonin and a lack of maternal circadian rhythm, which results in disruption of its synthesis in the placental tissue during pregnancy [87,135,136,137].

Such pathology is observed in women with chronic pathologies of three or more functional systems (cardiovascular, gastrointestinal, neural, etc.), obesity, diabetes mellitus, metabolic syndrome, endometriosis and polycystic ovary syndrome, as well as in women working at night [138,139,140,141,142,143]. It has been found that in such women pregnancies are complicated by chronic placental insufficiency, preeclampsia and gestational diabetes [144,145,146,147]. Numerous experimental and clinical studies provide evidence that in the case of preeclampsia there is no significant increase in epiphyseal melatonin (which is characteristic for noncomplicated pregnancies) and no melatonin circadian rhythm [88,139,148]. Reduced expression of melatonin receptors and melatonin production [140], as well as expressed oxidative stress [149] and increased levels of anti-inflammatory cytokines (Il-1b, TNF-a, Il-6), have also been observed [137].

Similar changes have been observed in pregnant women with pre- and gestational diabetes, hypertensive syndrome, obesity and in cases of smoking [150,151,152]. In the case of obesity and gestational diabetes, an alteration occurs in the expression of mRNA insulin receptors and glucose transporters (GLUT-1, GLUT-4) in adipose tissue and skeletal muscle, which significantly affects metabolic processes in the mother–placenta–fetus system [153]. Along with activation of free-radical oxidation, immune, trophic, endocrine and metabolic placental dysfunctions and chronic hypoxia have a damaging impact on the fetus [17,96,154].

This may result in delayed growth or fetal macrosomia, impairing the development of functional systems of the fetus, formation of interconnections between systems and circadian rhythm of their activities [116,135]. It is the fetus’s brain that suffers dramatically as it is most sensitive to oxidative stress and free-radical damage in comparison with other tissues [155]. Newborns with intrauterine growth retardation (IUGR) have an increased level of proinflammatory cytokines, as do their mothers who have had preeclampsia [156]. Disruption of maternal melatonin circadian production not only leads to delayed formation of the fetus’s specific genes’ rhythmic activity but also predetermines the endocrine pathology programming of the offspring [96,108,117,157,158].

## 6. Maternal Melatonin Deficiency and Endocrine Pathology Programming of the Offspring

In recent decades, numerous experimental and epidemiological studies have proved the role of epigenetic effects on various adverse factors in distorting the formation of regulatory mechanisms during critical development of the fetus’s organs and systems, which changes the nature of postnatal adaptive reactions in the new environment and contributes to a delayed development of pathological conditions.

Research into the mechanisms of the regulation of genome functions which are not associated with the DNA structure, i.e., epigenetic regulation, has established the effect of environmental factors on the activity of placental and fetal genes by changing DNA cytosine nucleotide methylation [159]. Covalent histone modifications (acetylation, methylation and phosphorylation) and activities of small regulatory and interfering RNAs also affect gene expression.

The frequency of epimutations due to these processes can exceed the frequency of gene mutations and cause mutability of phenotypic traits of the fetus. Without affecting DNA sequences, primary epimutations change the configuration of chromatin which can be either transcriptionally active or inactive. In that case, developmental pathology may be a consequence of the turning off (imprinting) of genes which normally should be functionally active, and reversely, a result of activation may occur due to hypomethylation of genes which in normal development should remain imprinted [154,160]. Gene-expression disorders are followed by changes in enzyme activity and dysregulation of cell differentiation and development, which eventually leads to a decrease in their number and irreversible morphofunctional disorders.

The environment and the genetic apparatus interact at all stages of placental and fetal development. In the case of hypoxia, in the placenta there is increased methylation in the promoter regions of the TUSC3 gene, and reversely hypomethylation of the p53 gene involved in apoptosis [126,161]. In newborns from mothers with obesity, diabetes and preeclampsia, there were changes in the expression of genes involved in the regulation of brain development, inflammation and immune signaling, carbohydrate and lipid homeostasis and oxidative stress [96,162,163,164].

Epigenetic changes due to the impact of adverse environmental factors predetermine consequences of perinatal damages. It is known that inflammatory changes in fetal adipose tissue and skeletal muscles of fetuses from mothers with obesity can cause excessive insulin production by the pancreas and insulin resistance [165]. Insulin receptors are highly expressed in the cortex and hippocampus, and synaptic insulin signaling plays a key role in learning and eating-behavior formation [166].

In an experimental study, in the case of maternal obesity and hyperinsulinemia, in the fetus’s hippocampus there was expression inhibition of insulin receptor genes and glucose transporters, as well as distorted development of the brain’s serotoninergic and dopaminergic systems involved in eating-behavior regulation [167,168]. In the case of chronic hypoxia, a change in expression of the epiphyseal AANAT (arylalkylamine N-acetyltransferase) gene can suppress the synthesis of melatonin [169]. Along with epigenetic mechanisms, hormonal mechanisms play an important role in being able to trigger epigenetic programming in the absence of melatonin [160].

## 7. Maternal Melatonin Prevents Pathology Programming

Various data obtained in recent decades regarding the mechanisms that determine adverse consequences have proved that only maternal melatonin protects the genetic development program of the mother–placenta–fetus systems from epigenetic damage. By controlling DNA methylation and histone modifications, melatonin prevents changes in the expression of genes that are directly related to the programming of various pathologies’ development [170,171,172,173,174]. As evident from experimental studies, such programming can be prevented or lessened by early administration of melatonin [155,175,176,177].

Thus, experimental studies on various animal models (rats and sheep) have established that, in the presence of adverse environmental factors, melatonin prevents the development of oxidative and nitrative stress in the placenta and fetus [119,136,178,179,180], suppresses production of proinflammatory cytokines and stimulates the production of anti-inflammatory cytokines in the maternal serum, amniotic fluid and fetal brain [181,182], stabilizes the blood-brain barrier and prevents inflammation development and neural apoptosis [183,184,185,186].

Melatonin maintains endoplasmic reticulum homeostasis, providing the protection of metabolic processes in stress conditions [187]. Melatonin administration to rats subjected to nicotine suppressed oxidative stress and reduced lung and liver damage in fetuses [188] and prevented neural-tube defects under hyperglycemia conditions in diabetes [189].

Antenatal melatonin administration in the case of placental insufficiency improved the function of the placenta, normalized the fetal–placental circulation, coronary blood flow, cardiac function and fetal growth [190,191,192]. Melatonin had multiple positive effects on mitochondria: it reduced the intensity of oxidative stress, maintained the mitochondrial membrane potential, increased the efficiency of ATP production, regulated an optimal balance between pro- and anti-apoptotic proteins, prevented the release of cytochrome C into the cytosol and inhibited caspase 3 activity [178,179,193].

The results of numerous experimental studies that proved the role of melatonin in optimal pregnancy outcomes in pathological conditions [194,195] have served as a basis for developing new approaches to its application in clinical obstetrics, particularly in cases of pregnancies complicated by preeclampsia and intrauterine growth retardation syndrome (Figure 1).

## 8. Conclusions

Thus, maternal melatonin deficiency and absence of circadian rhythm are behind distortions of genetically programmed development of hormonal and metabolic regulatory mechanisms in the child’s functional systems, which predetermines the programming and occurrence of endocrine pathology in early ontogenesis. The primary mechanisms forming this pathology are oxidative stress, mitochondrial damage, epigenetic regulation, glucocorticoid effect and actions of neuroactive steroids, somatolactogens and related peptides, in realization of which the absence or lack of melatonin promotes pathological progression in the following years. The importance of this factor in pathophysiological mechanisms of endocrine disorders determines the need for a new approach to risk assessment and timely prevention of offspring diseases already at the stage of family planning.

## Figures and Tables

**Figure 1 ijms-22-02058-f001:**
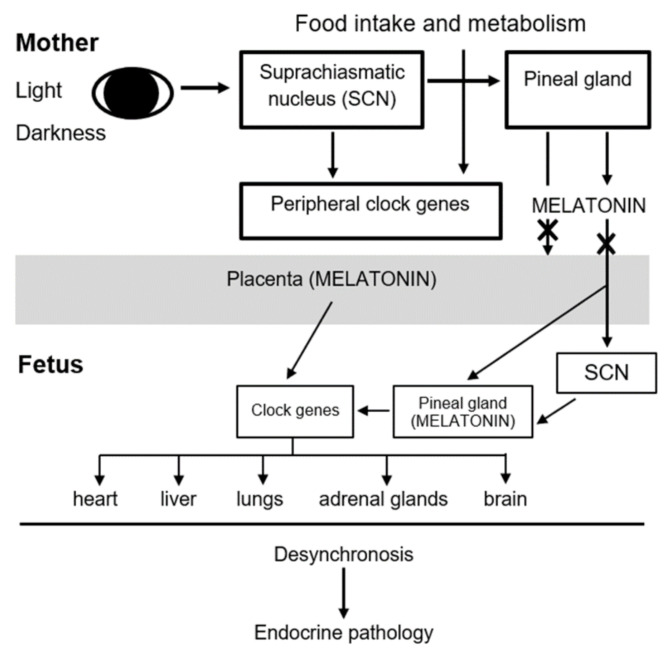
Maternal melatonin deficiency leads to endocrine pathologies in children.

## Data Availability

Data available in a publicly accessible repository.

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
