# Peer review of "Maternal Melatonin Deficiency Leads to Endocrine Pathologies in Children in Early Ontogenesis"

_ijms, 2021, doi:10.3390/ijms22042058_

Round 1

Reviewer 1 Report

The manuscript contains an extensive review of the experimental and clinical data that support the hypothesis of maternal melatonin deficiency having a role in endocrine pathologies in children in early ontogenesis. In general, the text is well addressed. It contains an abundant bibliographic description and is easy to read, although somewhere the organization can be improved or the text is not related to the tittle of the respective section. The authors claims often for a beneficial role of melatonin during the ontogenetic period what, at times, are tiring and repetitive. Likewise, there appear to be some mismatches between the text and the content of the referenced papers, which the authors should revised in detail. Here are some major points of my review.

It seems that overly ostentatious language is used throughout the manuscript to refer to the beneficial role of melatonin during early ontogenesis, and indeed the reader may sometimes get the feeling that melatonin is a miracle molecule. The document includes references to experimental and clinical data that unequivocally support the involvement of melatonin in circadian function, as well as its properties in the fight against oxidative stress. Therefore, the proposition that a melatonin deficiency at critical moments in ontogenetic development can have pathological consequences is really interesting. However, the authors frequently handle extrapolations from experimental results that are not supported by clinical investigations. These are usually studies with in vitro experimentation, in particular in relation to the antioxidant role of melatonin, as well as the effects produced by the application of exogenous melatonin, which can have different connotations. Furthermore, in many of these papers very specific processes are studied, some of which respond to melatonin (or its deficiency) and others not, which is not always taken into account in this review. The review would benefit if some of the hypothetical statements made are softened and the terms adjusted to the contents of the experimental studies.

The authors are recommended to review the allocation of bibliographic citations. In some referenced papers that I have been able to read, there does not seem to be a clear relationship between the data of these studies and what is mentioned in the text of this manuscript.

Pg 3, Ln 121-125. A role of melatonin in the control of endogenous rhythms in tissues, cells and subcellular levels is stated. I cannot agree with that proposal. It is clear that melatonin mediates the adjustment of rhythms in the body, but the control of rhythms itself depends primarily on the existence of a master clock or multiple secondary clocks in the body's cells and tissues.

Pg 4, first paragraph. The claim that melatonin triggers clock gene expression needs to be explained. In addition, it requires a bibliographic reference.

Pg 4, Ln 165-166. While the authors refer to very specific effects of melatonin at the metabolic level, they include references to reviews citing that research. Please reduce the citation to review articles and give more importance to specific research papers that focus on the topic.

Pg 4, Ln 181. MT1 receptors for insulin ?? Please check it

The citation from Varcoe et al., 2016 [80] should be revised. Perhaps, the interpretation of the results of this study is not correct. First, the experiments of this paper includes ClockΔ19+MEL mutant mice, but the pineal gland was not removed, as indicated in the text. Second, the reference to results of this paper seems to not adequately be addressed in the text. In fact, the study concludes that foetal growth within a genetically disrupted circadian environment during gestation has no effect on pregnancy success, and only marginal impacts upon the long-term metabolic health of offspring. Therefore, the dramatic effects noted in the present review are unclear.

Pg 4, Ln 204. The paper by Soliman et al. 2015 [88] should be revised. As far as I have been able to check, this study does not investigate the expression of clock genes in endothelial tissue.

Pg 5. The citation of Robera et al., 2008 ([83] does not seem to correspond to the text. Please check it. It should also indicated in which subjects the studies have been carried out, i.e., rodents in this study.

Pg 6. Ln263-266. I do not agree at all with the text. There is very little evidence that tryptophan content of breast milk and/or the inhibition of monoamine oxidase promoted by omega-3-poluinsaturated fatty acids increase the amount of serotonin and this causes an increase in melatonin levels. It is possible that these compounds affect the neurotransmitter content at the brain level, but I highly doubt that it has any effect on the serotonin content and melatonin synthesis in the pineal gland. In my view, the text is speculative and the conclusions go beyond the contributions of the mentioned research.

Section 5. "The state of placenta and fetal development in case of ..". The text in this section is quite disconnected. The problematic of free radicals oxidation is introduced without a connection to the previous topic. Then, some reference is made to melatonin similar to what was mentioned in previous points of the manuscript. The final part of the section is highly speculative, without clear contributions about the mechanisms that mediate the pathologies mentioned.

Section 6. "Maternal melatonin deficiency and endocrine pathology ...". There is no clear relationship between the title and the content of the section. References to studies that have used experimental and / or clinical models of melatonin deficiency are null. Furthermore, in this section the reference to melatonin is minimal.

Section 7. Several parts of the section report the results of experiments in which the effects of external, non-maternal melatonin have been evaluated. It would be interesting to mention the doses of melatonin used and the conditions in which the effects were assessed. From that, drawing open conclusions about a role for maternal melatonin seems risky. In my view it is too hypothetical and speculative.

Reviewer 2 Report

In this paper, Ivanov and colleagues reviewed the link between melatonin and children endocrine pathologies in early ontogenesis. They focused their attention on the mechanisms underlying the epigenetic influences during critical period of fetal organs development, particularly on the role of melatonin in regulating metabolic process. Indeed, melatonin, controls DNA methylation and histone modification, prevents changes in gene expression that are directly related to the programming of endocrine pathology in offspring. The authors finally pointed out that deficiency in maternal melatonin negatively determines the development of the hormonal and metabolic systems of the child.

The review is well conducted, but several revisions should be made prior its acceptance for publication:

A revision in the English grammar and syntax should be made throughout the text;

In the section 2 the authors should cite, as cause of hormonal alteration, the exposure to environmental toxicants, such as endocrine disrupting chemicals, also considering that in many studies melatonin has been used as a strategy to prevent their toxicity;

In the section 3, information concerning the signal transduction pathway activated by melatonin should be added;

Lines 197-198: the authors should better clarify what they meant with “increased circadian oscillations”.

Finally i suggestions to add some figures to render the review more suitable to the readers.

Round 2

Reviewer 1 Report

I have no further commens to add